# Students' Innovation in Education for Sustainable Development—A Longitudinal Study on Interdisciplinary vs. Monodisciplinary Learning

**Mirjam Braßler** [1,*] and **Martin Schultze** [2,*]

1. Institute of Psychology, University of Hamburg, 20146 Hamburg, Germany
2. Institute of Psychology, Goethe University Frankfurt am Main, 60629 Frankfurt am Main, Germany
* Correspondence: mirjam.brassler@uni-hamburg.de (M.B.); schultze@psych.uni-frankfurt.de (M.S.)

**Abstract:** Innovative ideas are essential to sustainable development. Students' innovative potential in higher education for sustainable development (HESD) has so far been neglected. Innovation is often associated with an interdisciplinary approach. However, the results of research on diversity and its role in innovation are inconsistent. The present study takes a longitudinal approach to investigating student teams in project-based learning courses in HESD in Germany. This study examines how innovation develops in interdisciplinary student teams in contrast to monodisciplinary student teams. The results of the latent change approach from a sample of 69 student teams indicate significant changes in students' innovation over time. Monodisciplinary student teams outperform interdisciplinary student teams in idea promotion (convincing potential allies) at the beginning, whereas interdisciplinary student teams outperform monodisciplinary student teams in idea generation (production of novel and useful ideas) in the midterm. There is no difference in the long term. The results indicate that interdisciplinary student teams have an advantage in the generation of novel ideas but need time to leverage their access to different discipline-based knowledge. We discuss practical implications for the design of interdisciplinary learning with strategies to support students in the formation phase in project-based learning in HESD.

**Keywords:** higher education for sustainable development; project-based learning; interdisciplinary learning; interdisciplinarity; student innovation

## 1. Introduction

Innovation is essential to ensure global sustainable development (SD). Enabling students to create change for sustainability is an unemployed resource in higher education for sustainable development (HESD) [1]. Both the generation of novel and useful ideas and the promotion of those ideas to gather alliances and coalitions for change, might be accessible to not only sustainability researchers and practitioners, but also the next generation: students. So far, students' innovative capacity has been neglected in HESD. Few studies have investigated the status quo of students' creativity within the field of SD [2,3] or their creativity skill development through HESD [4–6] or evaluated educational settings regarding their potential to enhance creativity in HESD [7–9]. The present study takes initial steps toward investigating students' innovation in HESD.

The 2030 Agenda for Sustainable Development includes global problems, such as ending poverty and hunger, climate change, protecting the planet from degradation, securing prosperity, and fostering peace [10]. Due to their complexity, these problems cannot be solved within one academic discipline [11–13]. To enable students to look for relationships, interactions, and possibilities to integrate different perspectives to generate holistic ideas, many researchers call for an implementation of interdisciplinary learning in HESD [14–18]. Due to their access to different discipline-based knowledge and methods, interdisciplinary student teams might be more innovative than monodisciplinary student

teams. Since both monodisciplinary [4] and interdisciplinary project-based learning [5] enhance students' creativity skills, a comparison of both pedagogies might add insight into students' innovation. Because research into team diversity and innovation has yielded mixed results and has mostly been limited to studies with a cross-sectional design [19], the present study takes a longitudinal approach to investigating whether students are more innovative in interdisciplinary project-based learning or in monodisciplinary project-based learning and how their innovation develops over time. As the first explorative study on students' innovation in HESD, this research strongly contributes to the understanding of students' idea generation and idea promotion in interdisciplinary and monodisciplinary teamwork. Using a longitudinal approach to analyze four occasions, the present paper adds insight into student team processes in project-based learning and provides practical implications for instructional design in interdisciplinary teaching and learning in HESD.

## 2. Theoretical and Empirical Framework

The present study draws on several theoretical perspectives. First, we give an overview of the constructs of creativity and innovation and previous research on students' potential in HESD. Second, we address educational settings—interdisciplinary and monodisciplinary project-based learning—and their potential to foster students' innovation in HESD. Third, we address the role of time in learning and innovation in students' teamwork in HESD. Finally, we integrate all theoretical perspectives by unfolding the research questions connecting students' innovation within the different educational settings over time.

### 2.1. Students' Creativity and Innovation in Education for Sustainable Development

The 2030 Agenda for Sustainable Development includes global problems, such as ending poverty and hunger, climate change, protecting the planet from degradation, securing prosperity and fostering peace [10]. All these problems are extraordinarily complex. They have many interdependencies, are multicausal, and often include conflicting goals. Levin Cashore, Bernstein, and Auld [20] define sustainability as "a super wicked problem" that triggers many problems simultaneously. To deal with the inherent complexity of these sustainability problems, capabilities for innovation are required [21]. Education for sustainable development (ESD) aims for students to become potential creative problem solvers regarding SD: "Young people must be recognized as one of the key actors in addressing sustainability challenges and be mobilized on key decision-making processes concerning sustainable development. Creative and innovative minds are among their strengths, and activities for young people should tap into these" [22] (p. 9).

Creativity is defined as "imaginative processes with outcomes that are original and of value" [23] (p. 118). A potential result of a creative process is innovation. Innovation refers to the introduction or application of novel ideas, processes, or procedures within a team that benefit individual, team, organization, or wider society [24]. Innovation includes the generation, promotion, and implementation of beneficial ideas [25]. Innovation begins with idea generation, that is, the production of novel and useful ideas in any domain [26]. The next task of the innovation process consists of idea promotion to potential allies such as team members, friends, backers, and sponsors. The promotion can result in the establishment of a coalition of supporters who provide the necessary power to realize the idea [27]. To enable students to approach and solve complex sustainability problems, creativity and innovation must be addressed in HESD.

Research into students' creativity and innovation in HESD is rather limited [1,3,7]. Most recent research addresses either the status quo of students' creativity within the field of SD or their creativity skill development through HESD or evaluates educational settings regarding their potential to enhance creativity in HESD. Students' innovative capacity in HESD has so far been neglected. Amran, Perkase, Satriawan, Jasin, and Irwansyah [2] investigated the status quo of students' 21st-century skill development in the context of SD. Within their approach to assessing 21st-century skills, they examined creative thinking as identifying problems, generating ideas, thinking divergently, using fluency, flexibility,

originality, and elaboration, and solving problems. Their results indicate that students had low levels of creative thinking in the context of SD. Cheng [3] examined the views of students on human creativity and environmental sustainability. The results of four case studies show that students had quite diverse views on creativity and sustainability. Students with experience in creativity and/or authentic experiences in sustainability used their creativity in daily life and self-discovery and supported the integration of creativity in ESD; however, students with little or no experience did not see the need to connect creativity and sustainability or support connecting creativity and ESD. Both studies point to the importance of implementing opportunities to enhance students' creativity and thereby activate their potential to generate innovation in HESD.

Few studies have explored students' creativity skill development through HESD. Heidt [4] analyzed business students participating in a project-based learning ESD course with a pre- and post-project reflection. The results reveal that students' skills in thinking divergently, risk taking, and overall creativity developed further. However, the results also demonstrate that students struggled with their development in seeking opportunities, tolerating ambiguity, as well as evaluating different ideas. Khandakar, Chowdhury, Gonzales, Touati, Emadi, and Ayari [5] studied electrical engineering students participating in a multi-course interdisciplinary project-based learning ESD approach and found that almost all students were strongly convinced that their participation enhanced their creativity. Carbonell-Carrera, Saorin, Melian-Diaz, and Torre-Cantero [6] evaluated a creativity ESD workshop by analyzing engineering students' creative skill development with a pre- and post-test of performance in the individual task of solving a geometrical problem in sustainable modeling. The results indicate significant gains in the following creativity skills: abreaction, originality, figurative expansion, expressive richness, and graphic skill. Enhancing students' creativity skills is fundamental to qualifying future sustainability problem solvers. However, these studies failed to consider students' application of creativity skills to generate innovation regarding SD.

In addition to the gap in the literature outlined above, few studies have investigated different educational settings regarding their potential to foster the application of creativity in HESD. Many theoretical and conceptual research approaches conclude that learning for innovation requires a learner-centered instead of a teacher-centered approach [1,28–33]. Further, students should be active and self-directed and work in collaborative groups on complex and real-world problems. This should occur within a production-based curriculum, with assessment focusing on creative and functional competencies. Consequently, suitable pedagogies to foster students' innovation capacity are problem-based learning, project-based learning, experiential learning, and inquiry learning. Zhou [9] used a mixed-method approach to study the drivers of and barriers to fostering a creative climate in project-based learning HESD in China. The results show that drivers of a creative climate include the challenge of the task, openness, trust between peers, experts' help, and group diversity regarding expertise. The results also reveal that group problems, such as poor project management, a lack of supervision support, ineffective communication, and students' risk aversion, are barriers to a creative climate in HESD. As both monodisciplinary [4] and interdisciplinary project-based learning [5] enhance students' creativity skills, both pedagogies might foster students' application of creativity to generate innovation in student teams in HESD.

Following Zhou's [9] results, group diversity should function as a driver of a creative climate and therefore might support students' innovation—idea generation and promotion—in project-based learning in HESD. Research into diversity suggests that its role in innovation might be more complex, which we address in the following section by comparing advantages and disadvantages of both pedagogies interdisciplinary and monodisciplinary project-based learning.



### 2.2. Interdisciplinary vs. Monodisciplinary Project-Based Learning in ESD

Project-based learning is one of the most popular teaching-learning formats in HESD [18,34,35]. Project-based learning was first introduced by Kilpatrick [36] as the project method to engage students in hearty and purposeful activities and was further developed by Blumenfeld et al. [37]. Project-based learning is defined as a pedagogy that entails two components: "a question or problem that serves to organize and drive activities; and these activities result in a series of artifacts or products, that culminate in a final product that addresses the driving question" [37] (p. 371). The learning activities are organized around achieving a shared goal by emphasizing students' independence, self-direction, inquiry, and collaboration, providing an authentic application of content and skills, and focusing on open-ended questions, while aiming for the development of 21st century skills [38–40]. Moreover, the pedagogy is often associated with interdisciplinarity [39,40].

Interdisciplinary learning is defined as a process by which "learners integrate information, data, techniques, tools, perspectives, concepts, and/or theories from two or more disciplines to craft products, explain phenomena, or solve problems, in ways that would have been unlikely through single-disciplinary means" [41] (p. 289). Adding interdisciplinary learning to project-based learning in HESD means that the focus on the sustainability product entails the application of different information, data, techniques, tools, perspectives, and so forth toward an innovative and effective product.

Many researchers expect positive results regarding students' innovation with the implementation of interdisciplinary teaching and learning in HESD, including in terms of divergent thinking [7], creativity [9], tolerance of uncertainty, and complex problem-solving [42]. Cebrián and Junyent [43] explored student teachers' views on ESD, with the results indicating the necessity of establishing a dialogue between disciplines to address sustainability issues holistically (social, economic, and environmental dimensions) and generate innovation. In the scientific sustainability community interdisciplinarity is often seen as key to identify novel solutions and approaches to SD [44–49]. These assumptions originate in the positive relation of interdisciplinarity and innovation based on the cognitive resource, information/decision-making perspective on diversity and innovation [19,50]. According to this perspective, diverse teams have access to a variety of knowledge, skills, and abilities due to the diversity among team members. The more knowledge available to members of the team, the more ideas the team can generate [51]. Consequently, an interdisciplinary approach to SD has the potential advantage of facilitating combinations of different domains of expertise with a discipline-based variety of methods, theories, and content. In contrast to the positive effect of diversity on innovation propagated by the cognitive resource and information/decision-making perspective, the social categorization perspective proposes a negative one. Theories on social identity and self-categorization [52,53] hold that similarities and differences are used to categorize the self and others into groups. Consequently, an interdisciplinary approach to SD has the potential disadvantage of triggering categorization processes that form subgroups within the team (i.e., "us" and "them") and thereby prevent the team from working effectively. Research on diversity in teams has yielded findings that are inconsistent [19]. Meta-analyses as well as individual studies find either positive, negative, or no effects of team diversity on outcomes such as performance or innovation [54–56].

Both perspectives on diversity—positive and negative—are reported in interdisciplinary research for SD [44–47]. While the advantages of multiple perspectives on social, economic, and ecological dimensions of complex sustainability problems are described in many cases [49], research also points to interdisciplinary scientific teamwork being prone to conflict. Interdisciplinary conflict often begins with misunderstandings [57,58]. Each academic discipline has its own patterns, meanings, values, knowledge traditions, codes of conduct, and ways of interacting with society [59,60]. Many disciplines have their own jargon and terminology [57,58], making it cumbersome to find a shared definition of common themes or problems across disciplines [61], especially considering that each discipline has a different understanding and definition of sustainability and SD [62]. Thus, identifying an interdisciplinary theme in SD is challenging, as finding common ground is

a typical barrier in interdisciplinary cooperation [46,58,63]. Cairns et al. [44] investigated interdisciplinary research projects addressing intersections between the Sustainability Development Goals. Their results indicate that research teams configure themselves to navigate the terrain of interdisciplinary sustainability research with 'innovation' and learning opportunities. However, they also report on messiness, chaos, and conflict. They conclude that interdisciplinary research teams on sustainability work on the thin line between success and failure. Similar to researchers in interdisciplinary science projects, students also reported to experience conflict in interdisciplinary teamwork due to discipline-based differences [34,35,62,64–66]. Particularly in interdisciplinary HESD, students' varying backgrounds, individual knowledge limitations in other disciplines, and different definitions and approaches to issues in SD become apparent [62]. Further, students find the interdisciplinary setting disturbing, and even frightening, because it differs from the ways in which they are accustomed to learning in monodisciplinary courses [34]. In contrast, students are used to monodisciplinary learning. They are familiar with their peers and share the same discipline-based knowledge. This might lead to easier communication since they speak the same academic language. Consequently, monodisciplinary teams might have less access to diverse knowledge to generate innovation but might have advantages in communicating and promoting their ideas to their team members with the same academic background. Getting acquainted with team members takes time, as does generating ideas regarding SD in project-based learning. Therefore, we address the topic of team development and the function of time in students' teamwork in the following section.

### 2.3. Interdisciplinary vs. Monodisciplinary Student Teamwork over Time

Teams in general [67,68], and particularly student teams in project-based learning in HESD [69] and interdisciplinary teams [70–72], are expected to form and develop over time. The most cited model on group development [73] is Tuckman's model of four (five) stages called forming, storming, norming, performing, and, in the revised model, adjourning [68,74]. After the establishment (forming), the group experiences conflict (storming), develops group cohesion (norming), works toward the accomplishment of a task (performing), and breaks up (adjourning). The Input–Process–Output model by Yasin and Rahman [69] also proposes a development in student teams in ESD. According to the model, student teams experience a phase of formation before identifying a problem and engaging in creative and innovative teamwork. Theories about interdisciplinary team development [70–72,75] propose that teams experience conflict and slowly evolve social and cognitive integration to perform successfully over time. Innovation is also not seen as the one-time result of momentum but rather as a continuous process of different cycles: recognition of opportunity, initiation, and employment [76,77]. A meta-analysis of more than 200 studies on work groups and teams in organizations concludes that teams are "complex, multilevel system(s) that function over time, tasks, and contexts" [78] (p. 517) and evolve through three major stages: forming, functioning, and finishing. The forming stage includes the development of trust, planning, and cognitive structuring. Functioning consists of bonding while managing the diversity of members and conflicts, adapting, and learning.

Based on these theoretical and empirical perspectives, both interdisciplinary and monodisciplinary student teams in project-based learning should evolve over time [69]. Aiming towards creating innovation, both types of teams should experience different phases of teamwork. Consequently, students' innovation should also change over time in both types of project-based learning.

### 2.4. Purpose of the Study

The purpose of this study is to investigate students' innovation in interdisciplinary and monodisciplinary learning in HESD. Interdisciplinary and monodisciplinary teams have different advantages and disadvantages regarding their innovative potential that might affect the students' teamwork at different points in time. Following theoretical

and empirical perspectives of team development, both types of student teams should evolve over time, albeit differently. To shed light into the development of students' work, we analyze two types of innovation: idea generation and idea promotion. The present study investigates whether students are more innovative in interdisciplinary project-based learning or in monodisciplinary project-based learning and how their innovation develops over the course of an entire semester. To this end, we investigate at the beginning of a semester-long course, after 4 weeks, 8 weeks and 12 weeks. Accordingly, the present study applies an explorative, longitudinal approach to investigate students' innovation in mono- and interdisciplinary learning in HESD.

## 3. Materials and Methods

### 3.1. Sample and Data Collection

The included HESD courses were selected because they have the following characteristics of project-based learning [38,64,79,80]:

- Duration: long-term project over a semester
- Problem: real-world, fully authentic tasks
- Process: follows the general, broad steps of project management—task analysis, identification of solutions, and implementation of solution
- Assessment: group assessment based on product

To identify courses meeting the criteria, the class schedules of five universities were screened. If the course titles included "sustainability" or "sustainable development" and the course descriptions indicated a monodisciplinary or interdisciplinary approach including teamwork, the instructors were contacted. In a call or in a meeting, the authors asked the instructors whether their courses met the criteria by going over each characteristic in the list. Most of the courses were excluded because their teaching methods did not include actual teamwork, or only included teamwork for a few sessions, so there was no continuous teamwork in teams of a constant composition over one semester. Further, most interdisciplinary courses were constructed with disciplines working in parallel (multidisciplinary) rather than being integrated (interdisciplinary) and therefore were not included in the sample.

The final sample consisted of 267 students (109 males, 40.8%, and 158 females, 59.2%) who were enrolled in an HESD course ($n$ = 12) at one of three higher education institutions in northern Germany. The mean age of the students was 24.59 years (SD = 4.22). Each student was a member of either one of 47 interdisciplinary teams or one of 22 monodisciplinary teams. All interdisciplinary student teams were assigned to integrate knowledge from all participating disciplines to generate innovation in HESD. The involved disciplines included psychology, economy, pedagogics, law, linguistics, physics, informatics, environmental studies, politics, geography, and mechanical engineering. All interdisciplinary student teams consisted of 2 to 3 different disciplines. All monodisciplinary teams consisted of psychology students who were also instructed to be innovative during their teamwork in their project-based learning course. All students remained in the same teams for the duration of the semester (a little more than 4 months). On four occasions during this period, each student was asked to fill out an instrument that took about 15 min. These occasions were after the first teamwork session, after one month, after two months, and after three months of student teamwork (see Figure 1).

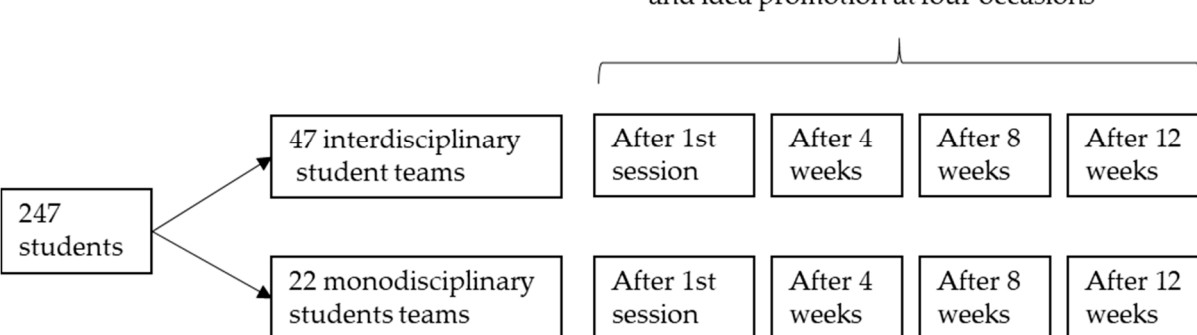

**Figure 1.** Operation Chart of Data Collection.

### 3.2. Measure

Students' innovative behaviors were measured with 6 items regarding initiation of innovation, 3 items on idea generation, and 3 items on idea promotion from the innovative work behavior (IWB) scale by Janssen [25]. The students were asked to rate their frequency of engaging in the following behaviors on a 5-point Likert scale:

- Creating new ideas for difficult issues (idea generation 1)
- Searching for new work methods, techniques, or instruments (idea generation 2)
- Generating original solutions for problems (idea generation 3)
- Mobilizing support for innovative ideas (idea promotion 1)
- Acquiring approval for innovative ideas (idea promotion 2)
- Making other team members enthusiastic about innovative ideas (idea promotion 3)

The last item was adapted to the educational context of the study by changing the original item "making important company members enthusiastic about innovative ideas" to "making other team members enthusiastic about innovative ideas." Following the translation and adaptation guidelines by Hambeleton and de Jong [81], all items were translated into German, then back into English, so that three native speakers could compare the original and backward translations in terms of literal and contextual equivalence with satisfying results (all over 80%).

### 3.3. Data Analysis

To investigate the development of students' innovative behaviors in HESD and compare it between inter- and monodisciplinary teams, responses to the IWB scale items were analyzed using a structural equation modeling (SEM) approach. We analyzed both latent constructs (idea generation and idea promotion) separately to reduce model complexity in light of the relatively low number of distinct university courses that were investigated ($n = 12$). In this study sampling was performed on the level of courses, introducing dependencies between the observations of students (i.e., students attending the same course are likely to be more similar to each other than they are to students from a different, randomly selected course). To account for these dependencies the approach proposed by Asparouhov and Muthén [82] for the correction of standard errors and fit statistics was used.

Because the focal point of this study is the development of students' innovative behaviors over time, we used the latent change approach. In this approach differences between latent constructs at different measurement occasions are incorporated as distinct latent variables. This allows for the prediction of change by independent variables such as the type of team a student worked in. In this study the latent change variables are computed as the difference between neighboring measurement occasions.

We dummy-coded the grouping variable ($G$) as follows that 0 indicates students from monodisciplinary teams and 1 indicates students from interdisciplinary teams. This results in a regression depicting the latent differences between neighboring occasions of

the monodisciplinary group in the intercepts ($\beta_0$) and the additional mean changes in the interdisciplinary group in the regression weights ($\beta_1$).

To ensure that the resulting parameters can be interpreted meaningfully, longitudinal measurement invariance was investigated using the step-by-step approach described by Widaman, Ferrer, and Conger [83].

## 4. Results

Sequential tests of measurement invariance using the strictly positive version of the Satorra-Bentler $\chi^2$ [84] revealed the model assuming strict measurement invariance to be best suited for idea generation, while the model assuming strong measurement invariance is determined to be best for idea promotion. Detailed results for the test of measurement invariance are provided in Table A1 of the Appendix A. In the case of idea generation, the final measurement model showed good overall model fit in terms of approximate model fit criteria (RMSEA = 0.042, SRMR = 0.065, CFI = 0.970), although the test of absolute model fit rejected the model ($\chi^2$ = 101.70, *df* = 69, *p* = 0.006). A similar pattern appeared for the model regarding idea promotion, where the approximate fit criteria indicated good model fit (RMSEA = 0.050, SRMR = 0.059, CFI = 0.966) and the test of absolute model fit was also significant ($\chi^2$ = 114.34, *df* = 69, *p* < 0.001).

Table 1 provides an overview of the means, reliabilities, and correlations of the latent states at the four measurement occasions as determined via the models with appropriate measurement invariance. The reliabilities in the assessment of change ranged from 0.630 to 0.717 for idea generation and from 0.800 to 0.818 for idea promotion.

The results of regressing the latent-change variables on the dummy-coded grouping variables are shown in Table 2. These show a significant decrease in idea generation in the monodisciplinary groups between the first two occasions ($\beta_0$ = −0.322, *p* = 0.025) and a significant difference between the two groups in this change ($\beta_1$ = 0.545, *p* = 0.001) in the form of interdisciplinary groups exhibiting a more positive development in terms of idea generation. Note that this also results in a significant mean increase from the first to the second occasion for the interdisciplinary groups ($\mu$ = 0.233, 95% CI: [0.091; 0.356], *p* = 0.001) as well as a significantly higher mean in these groups compared to the monodisciplinary groups at the second occasion ($\mu_{Inter}$ = −0.038, $\mu_{Mono}$ = −0.322, 95% CI of the mean difference: [0.076; 0.490], *p* = 0.007). Figure 2 shows the mean trajectories of both groups across the four occasions.

**Table 1.** Latent means, standard deviations, reliabilities, and correlations at each of the four occasions.

| | Latent Mean ($\mu$) | Standard Deviation | Reliability ($\rho$SEM) [1] | Correlations | | |
| --- | --- | --- | --- | --- | --- | --- |
| | | | | Occasion 1 | Occasion 2 | Occasion 3 |
| Occasion 1 | 0.00 [2] | 0.69 | 0.819 | | | |
| Occasion 2 | 0.06 | 0.70 | 0.823 | 0.448 | | |
| Occasion 3 | 0.00 | 0.71 | 0.827 | 0.411 | 0.638 | |
| Occasion 4 | −0.02 | 0.74 | 0.837 | 0.463 | 0.589 | 0.542 |
| Occasion 1 | 0.00 [2] | 0.91 | 0.878 | | | |
| Occasion 2 | 0.12 | 0.94 | 0.883 | 0.450 | | |
| Occasion 3 | 0.12 | 1.00 | 0.895 | 0.502 | 0.504 | |
| Occasion 4 | 0.05 | 1.05 | 0.904 | 0.330 | 0.579 | 0.501 |

Notes. [1] Reliability coefficient as presented by Yang and Green [85]. [2] The latent mean at the first occasion is fixed at 0 to identify the mean structure.

**Table 2.** Results from the regression predicting the latent changes in idea generation.

| | β | 95% Confidence Interval | Std. β [1] | p |
|---|---|---|---|---|
| | | State at T1 | | |
| Intercept [2] | 0.000 | - | - | - |
| Group Difference | −0.262 | [−0.639; 0.115] | −0.376 | 0.174 |
| | | Change between T1 and T2 | | |
| Intercept | −0.322 | [−0.602; −0.041] | −0.437 | 0.025 |
| Group Difference | 0.545 | [0.277; 0.863] | 0.740 | 0.001 |
| | | Change between T2 and T3 | | |
| Intercept | −0.016 | [−0.283; 0.207] | −0.026 | 0.891 |
| Group Difference | −0.054 | [−0.289; 0.181] | −0.090 | 0.651 |
| | | Change between T3 and T4 | | |
| Intercept | −0.004 | [−0.305; 0.296] | −0.006 | 0.977 |
| Group Difference | −0.056 | [−0.481; 0.370] | −0.079 | 0.798 |

Notes. The grouping variable is dummy-coded, with 0 indicating monodisciplinary and 1 indicating interdisciplinary groups. [1] Only the dependent variable is standardized. [2] The intercept at the first occasion is fixed at 0 to identify the mean structure.

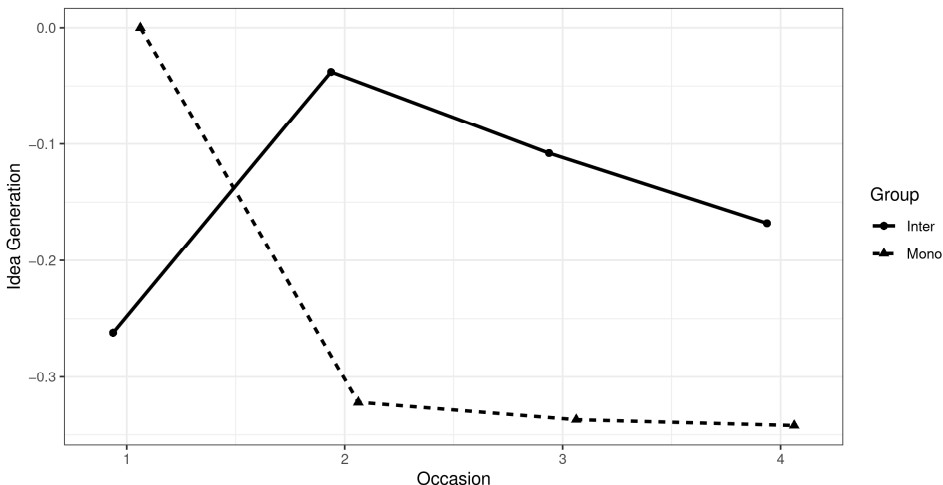

**Figure 2.** Mean trajectories of the interdisciplinary and monodisciplinary groups in idea generation.

Table 3 shows the results of the regressions predicting the latent changes in idea promotion. These indicate a significant difference between the group means at the first measurement occasion ($β_1 = −0.519$, $p < 0.001$), with the monodisciplinary groups having a higher latent mean. From the first to the second measurement occasion there is, however, a significant decrease in idea promotion in the monodisciplinary groups ($β_0 = −0.333$, $p = 0.012$). Additionally, the interdisciplinary groups show a more positive change between these two occasions ($β_1 = 0.645$, $p < 0.001$) than their monodisciplinary counterparts. Figure 3 shows the mean development across the four measurement occasions. Note that the significant mean decrease in the monodisciplinary groups between the first two occasions is accompanied by a significant mean increase in the interdisciplinary groups ($μ = 0.312$, 95% CI: [0.113; 0.511], $p = 0.010$). This results in the significant differences between the two groups at the first occasion disappearing at the second occasion ($μ_{Inter} = −0.206$, $μ_{Mono} = −0.333$, 95% CI of the mean difference: [−0.188; 0.440], $p = 0.509$). There are no notable differences between the two groups at later occasions.

**Table 3.** Results from the regression predicting the latent changes in idea promotion.

| | β | 95% Confidence Interval | Std. β [1] | *p* |
|---|---|---|---|---|
| | | State at T1 | | |
| Intercept [2] | 0.000 | - | - | - |
| Group Difference | −0.519 | [−0.688; −0.350] | −0.565 | <0.001 |
| | | Change between T1 and T2 | | |
| Intercept | −0.333 | [−0.550; −0.115] | −0.338 | 0.012 |
| Group Difference | 0.645 | [0.357; 0.932] | 0.655 | <0.001 |
| | | Change between T2 and T3 | | |
| Intercept | −0.071 | [−0.296; 0.155] | −0.073 | 0.606 |
| Group Difference | 0.099 | [−0.142; 0.341] | 0.102 | 0.499 |
| | | Change between T3 and T4 | | |
| Intercept | −0.098 | [−0.293; 0.098] | −0.093 | 0.410 |
| Group Difference | 0.026 | [−0.282; 0.335] | 0.025 | 0.889 |

Notes. The grouping variable is dummy-coded, with 0 indicating monodisciplinary and 1 indicating interdisciplinary groups. [1] Only the dependent variable is standardized. [2] The intercept at the first occasion is fixed at 0 to identify the mean structure.

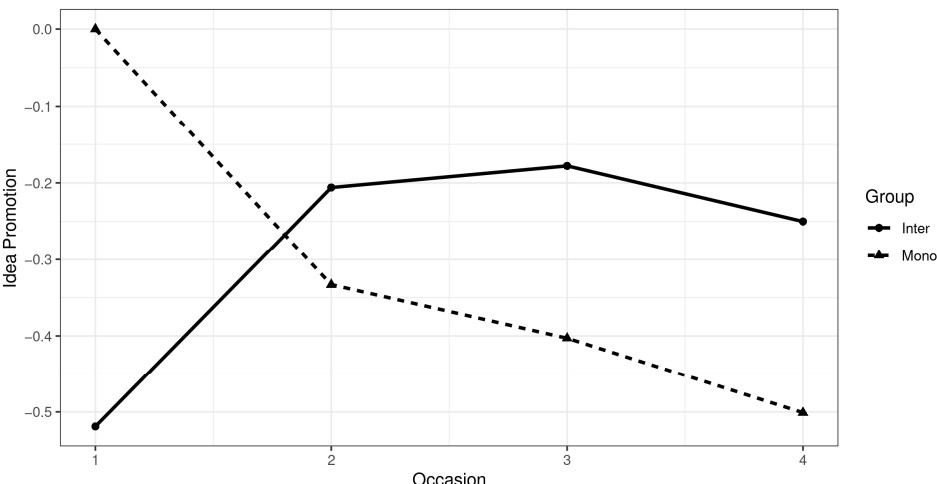

**Figure 3.** Mean trajectories of the interdisciplinary and monodisciplinary groups in idea promotion.

## 5. Discussion

Since novel and innovative ideas are essential to ensure global SD, there is a need to enable students to create change for sustainability. Students' creativity and innovation are still unemployed resources in HESD [1,9]. Few studies have investigated the status quo of students' creativity within the field of SD [2,3] or their creativity skill development through HESD [4–6] or evaluated educational settings regarding their potential to enhance creativity in HESD [7–9]. However, students' innovation has so far been neglected. The present study takes initial steps in investigating students' innovation in HESD. Innovation is highly associated with interdisciplinarity [86,87]. Due to their access to a variety of discipline-based knowledge, interdisciplinary teams are expected to be able to gain novel ideas to address complex sustainability problems [44–48]. However, research on team diversity yielded mixed results regarding innovation and is limited to cross-sectional research designs [19,50,54–56,88]. The present study's approach explores students' innovation over time in project-based learning—one of the most popular teaching–learning formats in

HESD [18,34,35]—to investigate whether students are more innovative in interdisciplinary or monodisciplinary learning.

The main result of this study points to students being more innovative in interdisciplinary learning in HESD, but this interdisciplinary advantage takes time. The results of the latent change models indicate that innovation in the form of idea promotion is higher in monodisciplinary student teams at the beginning, whereas there is no significant difference in innovative behavior in the form of idea generation between monodisciplinary and interdisciplinary student teams. Hence, students in monodisciplinary teams gather more support and approval for their novel ideas than students in interdisciplinary teams early on. In contrast, at the beginning of the teamwork, it makes no difference whether one is a member of an interdisciplinary or a monodisciplinary team in terms of gaining novel ideas, methods, and solutions. Focusing on teams' further development, the results of the latent change models indicate that interdisciplinary student teams outperform monodisciplinary student teams regarding idea generation in the medium term. In the long term, however, there is no difference between interdisciplinary and monodisciplinary student teams in idea generation. In idea promotion, interdisciplinary teams overcome their early disadvantage and catch up with the monodisciplinary teams in the midterm. Again, in the long term there is no difference between interdisciplinary teams and monodisciplinary teams in idea promotion.

Both interdisciplinary and monodisciplinary student teams experience changes in innovation over time. However, these changes are different. Monodisciplinary student teams experience a decrease in both idea generation and idea promotion after their first period of teamwork. In contrast, interdisciplinary student teams experience an increase in both idea generation and idea promotion after their first period of teamwork. Regarding team development, the beginning of teamwork in interdisciplinary project-based learning seems to be crucial. While monodisciplinary teams promote their novel ideas right from the start, interdisciplinary teams need time to make that significant jump and catch up. While monodisciplinary teams experience their personal peak of idea generation at the beginning, interdisciplinary teams outperform monodisciplinary teams in the midterm. In light of Tuckman's model of team development [68,74], interdisciplinary student teams need more time in the formation phase, indicating that students experience more insecurities and difficulties in identifying roles and common ground. According to Ilgen et al. [78], teams develop trust, start planning, and experience cognitive structuring in the forming stage. Since monodisciplinary team members are more similar in their work patterns [89] and approaches [90] and speak the same professional language [91], one can expect them to trust and understand each other more easily, thereby forming earlier as a team, enabling them to start working and being innovative. Due to their diversity, interdisciplinary teams must overcome those inhibitive differences at the beginning; in line with the results, this might require time. Supporting each other's ideas across disciplines, hence, idea promotion in interdisciplinary teams, appears to be more difficult than in monodisciplinary teams. Interdisciplinary teams take time to put to use their advantage of access to diversity, which is supported by the fact that they outperform monodisciplinary teams only after the initial phase. One can interpret the results as suggesting the initial domination of social categorization processes due to discipline-based differences in interdisciplinary student teams, explaining their shortfall in idea promotion in comparison to the monodisciplinary student teams. In the midterm, there might be a reduction in social categorization processes and the use of informational advantage, enhancing information/decision-making processes in interdisciplinary teams in idea generation. Since interdisciplinary student teams do not continue to outperform monodisciplinary student teams in the long term in either of the innovation facets, both factors, especially the social categorization processes, might still be intact. Slow development of social and cognitive integration in interdisciplinary teams is anticipated by several researchers [70–72]. Consequently, students need additional support at the beginning and further along in their teamwork in interdisciplinary problem-based learning. Hensley [7] points to the advantages of incorporating mindful-

ness to support learning and innovation in HESD. This might be especially beneficial in interdisciplinary learning. Paying attention to interdisciplinary communication as well as discipline-based differences might enhance students' active integration across disciplines right from the start. Zhou [9] identified poor management, a lack of supervisory support, and ineffective communication as typical barriers in project-based learning in HESD. These barriers might be even higher in interdisciplinary project-based learning, since interdisciplinary teams experience difficulties in identifying common ground, misunderstandings, and conflict [34,35,62,64–66]. The implementation of tutors who function as mediators might enhance understanding across disciplines at the beginning of interdisciplinary teamwork. Tutors—especially those in interdisciplinary team teaching—could provide interdisciplinary linkages, illustrate interdisciplinary possibilities, identify discipline-based misunderstandings, and give feedback on the interdisciplinary integration of content [64]. Further, students in both interdisciplinary and monodisciplinary project-based learning could benefit from creativity training [3] to enhance their innovative capacity.

The major strength of the present study is the longitudinal approach, with four measurement occasions investigating latent change in students' innovation in HESD. However, there are several limitations in the study that warrant further attention in future research. First, successful generation and promotion of ideas in project-based learning might depend on components other than only the educational model, constituting a missing variable bias. Potentially important variables could be students' personal characteristics and skills, such as openness, curiosity, creativity, and patience, and for interdisciplinary project-based learning, interdisciplinary experience [92].

Second, the questionnaires measuring students' innovation might have activated repeated self-reflections and therefore might have implicitly fulfilled the function of a reflection intervention. Since self-reflection and process reflection were identified as key to successful teamwork in diverse teams [93,94] and interdisciplinary collaboration [61,66,95], the application of the instrument might have facilitated innovation due to repeated measurement occasions.

Third, even though innovation research typically measures idea generation and idea promotion with self-report items [25], the use of a self-report inventory for collecting data may cast doubt on the validity of the measure. Future research might apply a direct measurement of innovation with the implementation of third-party ratings regarding novelty and usefulness. Similarly, measuring educators' appraisals might be beneficial, especially if they represent different academic disciplines and are therefore able to estimate novelty in interdisciplinary student products in interdisciplinary project-based learning.

Fourth, the present study is limited to the investigation of only psychology students in monodisciplinary project-based learning. Since the academic discipline of psychology is a discipline from the branch of social sciences and a relatively young discipline, future research might investigate different disciplines in monodisciplinary project-based learning with a wider range and variety of disciplines, especially, also including formal and natural sciences.

Finally, future research may investigate underlying mechanisms of innovation in interdisciplinary student teams. The combination of variables regarding social categorization processes as well as decision-making processes might be fruitful. To investigate change and interactions regarding these variables over time, future research should collect data from more student teams than the present study. Further, future research should address the connection between students' creativity skills and innovation in HESD.

This study has several practical implications. Monodisciplinary and interdisciplinary student teams function differently regarding innovation in HESD. The results indicate that interdisciplinary teams struggle in the formation phase. Consequently, both educators and student team members should be aware that innovation takes time in interdisciplinary project-based learning. Moreover, being patient and remaining calm pays off since interdisciplinary teams catch up in the midterm. To enhance interdisciplinary teams' innovation right from the start, educators could apply diversity training, focusing on typical interdisci-

plinary challenges and introducing tools to support creativity before the actual teamwork begins. Furthermore, the formation phase of interdisciplinary teamwork might benefit from professional mediators or tutors who moderate first meetings and resolve potential conflict.

**Author Contributions:** Conceptualization, M.B.; methodology, M.B. and M.S.; software, M.S.; validation, M.B. and M.S.; formal analysis, M.S.; investigation, M.B.; resources, M.B.; data curation, M.B.; writing—original draft preparation, M.B. and M.S.; writing—review and editing, M.B. and M.S.; visualization, M.S.; project administration, M.B.; funding acquisition, M.B. All authors have read and agreed to the published version of the manuscript.

**Funding:** This research received no external funding.

**Institutional Review Board Statement:** Ethical review and approval were waived for this study by the department of work and organizational psychology at the University of Hamburg.

**Informed Consent Statement:** Informed consent was obtained from all subjects involved in the study.

**Data Availability Statement:** Data as well as the Mplus syntax used for the analyses shown in this manuscript are available at https://osf.io/qe36t/.

**Conflicts of Interest:** The authors declare no conflict of interest.

## Appendix A

**Table A1.** Results of measurement invariance testing for the models of Idea Generation and Idea Promotion. The columns under Model Comparison denote the comparison of an invariance level with the previous one.

| | | | | | | | | Model Comparison | | | |
| Measurement Invariance | $\chi^2$ | $df$ | $p$ | RMSEA | SRMR | CFI | BIC | $\Delta\chi^2$ | $\Delta df$ | Correction Factor | $\Delta p$ |
|---|---|---|---|---|---|---|---|---|---|---|---|
| | | | | | Idea Generation | | | | | | |
| Configural | 82.22 | 48 | 0.002 | 0.052 | 0.053 | 0.969 | 5943.01 | - | - | - | - |
| Weak | 87.39 | 54 | 0.003 | 0.048 | 0.055 | 0.970 | 5912.81 | 2.27 | 6 | 1.47 | 0.893 |
| Strong | 90.98 | 60 | 0.006 | 0.044 | 0.056 | 0.972 | 5884.60 | 3.47 | 6 | 1.54 | 0.748 |
| Strict | 101.70 | 69 | 0.006 | 0.042 | 0.065 | 0.970 | 5847.06 | 7.49 | 9 | 1.71 | 0.586 |
| | | | | | Idea Promotion | | | | | | |
| Configural | 78.50 | 48 | 0.004 | 0.049 | 0.038 | 0.977 | 5711.42 | - | - | - | - |
| Weak | 91.60 | 54 | 0.001 | 0.051 | 0.048 | 0.972 | 5686.11 | 4.38 | 6 | 1.88 | 0.625 |
| Strong | 106.53 | 60 | <0.001 | 0.054 | 0.051 | 0.965 | 5664.31 | 5.57 | 6 | 2.11 | 0.473 |
| Strict | 114.34 | 69 | <0.001 | 0.050 | 0.059 | 0.966 | 5648.79 | 19.04 | 9 | 1.83 | 0.025 |

Note: *df*: Degrees of freedom, RMSEA: root mean squared error of approximation, SRMR: Standardized root mean residual, CFI: Comparative fit index, BIC: Bayesian Information Criterion, Correction factor indicates the necessary correction for the strictly positive Satorra-Bentler $\chi^2$.

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
