# Peer review of "Students’ Innovation in Education for Sustainable Development—A Longitudinal Study on Interdisciplinary vs. Monodisciplinary Learning"

_sustainability, doi:10.3390/su13031322_

Round 1
Reviewer 1 Report
The paper presents a wide literature review in Section 2 but it lacks of a clear statement of the research question with respect to the literature framework. I would appreciate a stricter link between the theorethical framework and the empirical analysis, which seem two separate bodies (two different authors?).
other concerns:
1) Paper tries to draw some general conclusions on differences between interidisciplinary and monodisciplinary educational teams but the monodisciplinary teams analysed include only psychology students. Conclusions should be referred only to monodisciplinary psychological students.
2) Data analysis description should be improved: it is disproportionally synthetic as compared to literature review. Figure is not intuitive for those not familiar with the latent change approach. For example, I'm puzzled by what the authors call occasions 1-4 in the figure and T 1-4 in the subsequent table. When symbols are introduced it should be defined their description (see also equation 1). A table with descriptive statistics could help.
Reviewer 2 Report
This work is of great interest from the point of view of research of innovations in education. From our point of view, the difference between monodisciplinary and interdisciplinary approaches is obvious and requires no proof. However, in terms of developing an interdisciplinary approach and popularizing it from a scientific point of view, this work is very important. It is very important, in our opinion, that the authors offer a review of the literature and existing solutions to the problem under consideration. In the main part of the study, the presented data analysis, accompanied by graphical representations of the results obtained, allows us to evaluate the work done in a comprehensive manner. The main advantage of the presented study is its longitudinal nature, which allows us to speak about the reliability of the results obtained. If we talk about what we lacked in this study, then we would present a General algorithm for the sequence of operations that formed the basis of the study, since this would make it possible to present the General logic of the study.
Round 2
Reviewer 1 Report
I find the revised version strongly improved in its coherence and readeability